# Sparse Signal Models for Data Augmentation in Deep Learning ATR

**Tushar Agarwal ***[ID], **Nithin Sugavanam and Emre Ertin**[ID]

Department of Electrical and Computer Engineering, The Ohio State University, Columbus, OH 43210, USA; sugavanam.3@osu.edu (N.S.); ertin.1@osu.edu (E.E.)

*   Correspondence: agarwal.270@osu.edu

**Abstract:** Automatic target recognition (ATR) algorithms are used to classify a given synthetic aperture radar (SAR) image into one of the known target classes by using the information gleaned from a set of training images that are available for each class. Recently, deep learning methods have been shown to achieve state-of-the-art classification accuracy if abundant training data are available, especially if they are sampled uniformly over the classes and in their poses. In this paper, we consider the ATR problem when a limited set of training images are available. We propose a data-augmentation approach to incorporate SAR domain knowledge and improve the generalization power of a data-intensive learning algorithm, such as a convolutional neural network (CNN). The proposed data-augmentation method employs a physics-inspired limited-persistence sparse modeling approach, which capitalizes on the commonly observed characteristics of wide-angle synthetic aperture radar (SAR) imagery. Specifically, we fit over-parametrized models of scattering to limited training data, and use the estimated models to synthesize new images at poses and sub-pixel translations that are not available in the given data in order to augment the limited training data. We exploit the sparsity of the scattering centers in the spatial domain and the smoothly varying structure of the scattering coefficients in the azimuthal domain to solve the ill-posed problem of the over-parametrized model fitting. The experimental results show that, for the training on the data-starved regions, the proposed method provides significant gains in the resulting ATR algorithm's generalization performance.

**Keywords:** machine learning; data augmentation; automatic target recognition; synthetic aperture radar

## 1. Introduction

Synthetic aperture radar (SAR) sensors provide day and night high-resolution imaging capabilities that are robust to weather and other environmental factors. The SAR sensor consists of a moving radar platform with a collocated receiver and transmitter that traverses a wide aperture in the azimuth domain, acquiring coherent measurements of scene reflectivity. The returns for multiple pulses across the synthesized aperture are combined and coherently processed to produce high-resolution SAR imagery. A SAR imaging system achieves a high spatial resolution in both the radial direction, termed as range, as well as in the orthogonal direction, termed as cross-range. The range resolution is a function of the bandwidth of the signal used in illumination. The cross-range resolution is a function of the antenna aperture's size and the persistence of scattering centers [1]. A significant fraction of the energy in the back-scattered signal from the scene is due to a small set of dominant scattering centers that are resolved by the SAR sensor. The localization of back-scatter energy provides a distinct description of the targets of interest [2], such as in the case of man-made objects such as civilian and military vehicles. This sparsity structure has been utilized in [3,4] to design features like peak locations and edges that succinctly represent the scene. In the early works, these hand-crafted features were used in solving the target recognition problem in a statistical framework. Notably, the template-based methods exploited

the geometric structure and variability of these features in the scattering centers in [5,6] to distinguish between the different target categories. The target signature of each of the scattering centers varied with the viewing angle of the sensor platform. Statistical methods can explicitly model and utilize this low-dimensional manifold structure of the scattering center descriptors [7,8] for improved decisions, as well as for integrating information across views [9,10].

However, ATR algorithms based on these hand-crafted features are limited to the information present in these descriptors, and they lack the generalization ability with respect to variability in clutter, pose, and noise. With the advent of data-driven algorithms such as artificial neural networks (ANN) [11], an appropriate feature set and a discriminating function can be jointly estimated using a unified objective function. Recent advances in techniques to incorporate the deep hierarchical structures used in ANN [12,13] has led to the widespread use of these methods to solve inference problems in a diverse set of application areas. Convolutional neural networks (CNN) in particular have been used as automatic feature extractors for image data. These methods have also been adopted in solving the ATR problem when using SAR images [14]. There have been several efforts in this direction, including the state-of-the-art ATR results in the MSTAR data set in [15]. These results establish that a CNN could be effective in radar image classification when provided with sufficient training data. However, this approach of designing ATR algorithms for new sensors that operate in different bands and elevations with limited training data from targets of interest is not feasible as the scattering behavior changes substantially as the wavelength of the operation changes. The major challenge is that neural networks usually require large data sets to have a good generalization performance. In general, labeled radar image data are not readily available in abundance unlike other image data sets. In this paper, we address the scarcity of training data and provide a general method that utilizes a model-based approach to capture and exploit the underlying scattering phenomenon to enrich a training data set.

Transfer learning is one of the most effective techniques through which to handle the availability of limited training data. Transfer learning uses the model parameters, which are estimated using a similar data set such as Image-net [16], as initialization for solving the problem of interest, and typically CNNs are used with little to no fine-tuning. There have been numerous experiments supporting the benefits of transfer learning, including two seminal papers [17,18]. However, radar images are significantly different from regular optical images. In particular, SAR works in the wavelength of 1 cm to 10 m, while visible light has a wavelength of the order of 1 nm. As a result, most surfaces in natural scenes are rough at visible wavelengths, leading to diffused reflections. In contrast, microwaves from radar transmitters undergo specular reflections. This difference in scattering behavior leads to substantially different images in SAR and optical imaging. Since specular reflections dominate the scattering phenomenon, the images are sensitive to instantaneous factors like the imaging device's orientation and background clutter. Therefore, readily available optical-imagery based deep neural network models like Alex-net and VGG16 [19] are not suitable for transferring knowledge to the SAR domain. In this paper, we pursue an alternative strategy for the data augmentation of limited data sets through using a principled approach that exploits the phenomenology of the RF backscatter data.

Next, we review the relevant research work and outline our contributions in Sections 1.1 and 1.2, respectively.

## 1.1. Related Work

Over-fitting is a modeling error that is common to data-driven machine learning methods when the learned classifier function is too closely aligned to the training data points and therefore fails to generalize to the data points outside the support of the training set. The over-fitting problem is exacerbated with smaller training sets. Several methods have been proposed to reduce over-fitting and to improve the generalization performance. Typically, the ill-posed problem of fitting an over-parametrized function to data is solved

by using regularizers that impose structure and constraints in the solution space. The norm of the model parameters serves as a standard regularizing function. This keeps the parameter values small with a 2-norm ($|| \cdot ||_2$ called $L_2$ loss) space or sparse with a 1-norm ($|| \cdot ||_1$ called $L_1$ loss) space. Furthermore, the optimization algorithms, such as stochastic gradient descent and mirror descent, implicitly induce regularization [20,21]. Dropout, introduced by Srivastava et al. [22], is another popular method specifically for deep neural networks. The idea is to randomly switch off certain neurons in the network by multiplying a Bernoulli random variable with a predefined probability distribution. The overall model learned is an average of these sub-models, providing improved generalization performance. Batch normalization is another way through which to improve the generalization performance, and was proposed by Ioffe and Szegedy [23]. They proposed normalizing all the neuron values of the designated layers continuously while training them along with an adaptive mean and variance that would also be learned as part of the back-propagation training regime. Finally, the work by Neyshabur et al. [24] established the benefit of over-parameterizing in implicitly regularizing the optimization problem and in improving the generalization performance.

Transfer learning is another approach for improving the generalization performance in the cases where there is a limited availability of data. Pan and Yang [25] provided a comprehensive overview that illustrated the different applications and performance gains of transfer learning. For radar data, Huang et al. [26] suggested a promising approach along this direction by using a large corpus of SAR data to train feature extractors in an unsupervised manner. Huang et al. [27] recently extended this idea to high-resolution SAR data.

Much of the recent literature has gravitated towards using CNNs as the automatic feature extractors for the SAR ATR task, as mentioned earlier. One research direction has been to improve the classifier of CNNs by cascading CNN feature extraction with other machine learning algorithms, such as a large-margin softmax classifier in [28] and an ensemble learning-based classifier called the AdaBoost rotation forest in [29]. Other researchers have focused on improving ATR performance through multiple views of the same object as in [30], or by using multiple polarization information as in [31,32]. There is a great deal of potential in improving ATR performance, especially in challenging scenarios such as clutter. In such scenarios, polarimetric data and bistatic measurements serve as an important tool in improving the classification performance, especially in cases of low sample sizes in the training data. Our phase history model works with complex-valued data, and we previously extended the model to bistatic measurements in [33]. Another important research direction has been focused on reducing the space and computation requirements of CNN-based ATR. To achieve this, depthwise separable convolutions were used in [34]; in addition, Huffman coding and weight quantization were used in [35], as well as knowledge distillation in [36,37]. Few other approaches focus on learning a special type of features. Dong et al. [38] generated an augmented monogenic feature vector followed by a sparse representation-based classification. In [39], the authors used hand-designed features with supervised discriminative dictionary learning to perform SAR ATR. Song et al. used a sparse-representation-based classification (SRC) approach in [40]. In [41], Huang et al. designed a joint low rank and sparse dictionary to denoise the radar image while keeping the main texture of the targets. Yu et al. [42] proposed a combination of Gabor features and the features extracted by neural networks for better classification performance.

When there is a limited availability of SAR data, there exist several ANN-architecture-based approaches to improve generalization. Chen et al. [14] restricted the effective degrees of freedom of a network by using a fully convolutional network. Lin et al. [43] proposed a convolutional highway network to tackle the problem of limited data availability. In [44], the authors designed a specialized ResNet architecture that learns effectively even when the training data set is small. In addition to [26], the idea of semi-supervised learning has recently received much attention for improving SAR ATR performance in cases of limited

data availability. Yue et al. [45] used a CNN to obtain the class probabilities of unlabeled data samples, which was followed by incorporating this knowledge into the classification loss of ATR via a novel linear discriminant analysis method. In [46], Wang et al. used the information in unlabeled SAR data to inform a deep classifier by using a self-consistent augmentation rule, a mixup-based mixture, and weighted loss. Recently, Chen et al. [47] used an unlabeled data-based consistency criterion, domain adaptation, and top-k loss to alleviate the requirement of labeled data.

Data augmentation is another example of a regularization strategy that reduces the generalization error while not affecting the training error [48,49], and is the main focus of this paper. The main idea is to use domain-specific transformations to augment the original training data set. J. Ding et al. [50] explored the effectiveness of the conventional transformations used for optical images, viz. translations, noise addition, and linear interpolation (for pose synthesis). They reported marginal improvements in classification performance on the MSTAR data set. Yan [51] used the original training images to generate noisy samples at different signal-to-noise ratios, multiresolution representations, and as partially occluded images. In [52], the authors proposed a generative adversarial network (GAN) to generate synthetic samples for the augmentation of SAR data, but they did not report any significant improvements in the error rate of the ATR task. Lewis et al. [53] explored multiple deep generative models for SAR data augmentation and recommend BicycleGAN after experimentation. In another effort that used a GAN, Gao et al. [54] used two jointly trained discriminators with a non-conventional architecture. They further used the trained generator to augment the base data set and reported significant improvements. Cui et al. [55] used a Wasserstein GAN, and Sun et al. [56] proposed an attribute-driven angular rotation generative network to produce synthetic samples for augmentation. Shi et al. [57] used a GAN to super-resolve samples for data augmentation. None of these deep generative methods used complex-valued imagery; therefore—unlike the proposed work here—they were unable to create imagery that was consistent with the frequency support of the imaging system. Cha et al. [58] used images from a SAR data simulator and refined them using a learned function from real images. Simple rotations of radar images were considered as a data augmentation method in [59]. In [15], Zhong et al. suggested key ideas for incorporating prior knowledge in training the model. They added samples that were flipped in the cross-range dimension with a reversed sign of the azimuthal angle. Such flip-augmentation exploits the symmetric nature of most objects in the MSTAR data set. They also added a loss that was auxillary to the primary objective of classification. The authors used the pose prediction (azimuthal angle) as the secondary objective of the network. They empirically showed that this helps by adding meaningful constraints to the network learning. Thus, the network was more informed about the auxiliary confounding factor, improving its generalization capability. Lv and Liu [60] proposed to extract attributed scattering centers (ASCs) through the sparse representation algorithm. The synthetic samples for data augmentation were then reconstructed by selecting a subset of these ASCs and by repeating the procedure.

### 1.2. Contributions

In this work, we introduce a novel data augmentation method for SAR domains, following a principled approach that exploits the phenomenology of the RF backscatter data over the azimuth and frequency domains. This paper is an extension of our previous work [61] with additional results that include a comparison to other existing techniques and an ablation study of the components of the proposed technique.

First, we introduce an approach for pose synthesis that models and exploits the limited persistence of the sparse set of scatterers over the azimuth domain. We assume that man-made objects comprise a small set of dominant scattering centers. Specifically, we first transform the image into the polar frequency domain to obtain the samples in the phase history domain. We then construct a model motivated by the scattering behavior of canonical reflectors in this phase history domain. The phase history model further decouples the point-spread function associated with the imaging setup. This model captures the

phenomenology of the viewing-angle-dependent anisotropic scattering behavior of man made-objects, as well as provides realistic imagery at poses outside the training data set, with quality that far surpasses previous approaches such as linear interpolation in image domains [50].

Second, with modeling in the complex valued phase history domain, our algorithm can create realistic sub-pixel shift augmentations that capture the well-known scintillation effects in SAR imagery. These sub-pixel shifts are not possible in traditional image domains that use standard interpolators (linear, cubic, etc.), as the complex-valued interpolation kernels need to be appropriately designed by taking into account the azimuth and frequency windows of the sensor. We hypothesize that these two factors are essential for improving the network's knowledge about the SAR imaging systems' underlying physics.

Third, we focus on a state-of-the-art deep learning classifier for SAR ATR [15] using the MSTAR data set, as well as provide extensive simulation studies to illustrate the learning performance of different training data set sizes with un-augmented and augmented approaches to training. Our results show a significant boost in the generalization performance over both un-augmented and augmentation approaches with the previously suggested approaches. In particular, for the MSTAR data set when the training data set is reduced by a factor of 32, the proposed augmentation algorithm reduces the test error by more than 42% when compared to the baseline approach that includes image domain flips and integer pixel translations.

It is important to note that our data-augmentation-based strategy is generic and decoupled from the network architectures proposed in other works like [14,43]. Therefore, the proposed augmentation strategy may yield even further improvements in conjunction with the methods mentioned above. Our objective here is to demonstrate the benefits of the proposed data augmentation strategy. Hence, apart from data augmentation, we only use Zhong et al.'s [15] multi-task learning paradigm.

The rest of the paper is structured as follows. In Sections 3.1 and 3.2, we describe the data set and network architecture in detail. In Section 2.1, we provide an overview of our strategy and then describe the details of our pose-synthesis methodology in Section 2.2. Following those sections, we present the details of the experiments and corresponding results in Sections 3 and 4, respectively, which provide the empirical evidence for the effectiveness of the proposed data augmentation method. We then conclude with some possible directions for future research in Section 5.

## 2. Model-Based SAR Data Augmentation

An approach to ATR algorithm design is to train a parametric neural network classifier $g$, with parameters $w \in \mathbb{R}^{d_w}$, that predicts an estimate of output labels $Y \in \mathbb{R}^{d_Y}$ for an input $X \in \mathbb{C}^{d_X}$, i.e., $\hat{Y} = g(X; w)$, where $d_X$, $d_w$, and $d_Y$ are dimensions of $X$, $w$, and $Y$, respectively. We consider a supervised learning setting, where a labeled training data set $\mathcal{D}_{train} = \{(X_u, Y_u)\}_{u=1}^{N_{train}}$ is used to estimate the classifier parameters $w$, where $N_{train}$ is the total number of training samples. The training procedure is the minimization of an appropriate loss function $\mathcal{L} : (w, \mathcal{D}) \to \mathbb{R}$, which is achieved by using an iterative algorithm like the stochastic gradient descent. Therefore, the learned $w^*$ is the solution of the following minimization problem $\mathcal{P}$:

$$w^* = \mathcal{P}(\mathcal{D}) = \arg \min_w \mathcal{L}(w, \mathcal{D}) \tag{1}$$

Data augmentation involves applying an appropriate transformation, such as $T\mathcal{D}_{in} \to \mathcal{D}_{out}$, to a data set (only using $\mathcal{D}_{train}$ for our purposes), and to then expand it to an augmented data set $T(\mathcal{D}_{train})$. We also use a validation data set, $\mathcal{D}_{val} = \{(X_u, Y_u)\}_{u=1}^{N_{val}}$, for cross-validation during training. Furthermore, we use a test data set $\mathcal{D}_{test} = \{(X_u, Y_u)\}_{u=1}^{N_{test}}$ for evaluating $g(X; w)$ in post-training. The evaluation can be conducted using a suitable metric $\mathcal{M} : (w, \mathcal{D}) \to \mathbb{R}$, which may be different from the $\mathcal{L}$ above. Our aim is to find $T$, such that the estimated parameters $w_{aug} = \mathcal{P}(T(\mathcal{D}_{train}))$ perform better than

$w_{train} = \mathcal{P}(\mathcal{D}_{train})$ in terms of the chosen metric, i.e., $\mathcal{M}(w_{aug}, \mathcal{D}_{test})$ is more desirable than $\mathcal{M}(w_{train}, \mathcal{D}_{test})$.

### 2.1. Exploiting SAR Phenomenology for Data Augmentation

Simple approaches to designing transformations $T$ for data augmentation consider translation invariance and the symmetry of objects around its main axis in order to introduce discrete pixel shifts and flips along the cross-range dimension. Our augmentation strategy goes further and uses model-based transformations to improve the network's knowledge principally about two confounding factors, the pose as well as the scintillation effects that occur due to shifts in the range domain. Our method allows the synthesis of new poses in a close neighborhood of existing poses, based on a sparse modeling of the existing training data set that exploits the spatial sparsity and the scattering centers' limited persistence.

An overview of our approach is as follows: for every image in the training data set, we fit a sparse model in the phase history domain by exploiting SAR phenomenology. This model is henceforth referred as the PH model. We utilize the continuity of this PH model in the azimuth domain to extrapolate the phase history measurements and to synthesize new images in a close neighborhood of the original image. The PH model also allows for the introduction of arbitrary-valued sub-pixel shifts in both range and cross-range dimensions to images at both the original and synthesized poses. These fractional shifts provide information to the network regarding scintillation effects, which further improves its generalization capability. In the following section, we describe our modeling and pose synthesis strategy in full detail.

### 2.2. Modeling and Pose Synthesis Methodology

This section describes the pose synthesis methodology used for data augmentation when using the PH model. This work builds on our earlier work, which focused on modeling of the scattering behavior of targets in monostatic and bistatic setups [33,62–68]. We first constructed a model for each image in the training data set and locally extrapolated the measured images through using the model. We assumed that a SAR sensor that operated in the spotlight mode was used to create the images (as in the case of the MSTAR data set). The images were translated from the spatial domain to the Cartesian frequency domain via the steps described in [69]. Subsequently, we converted the frequency measurements to the polar coordinates to obtain the phase history measurements described in [70].

We considered a square patch on the ground of side lengths $L = 30$ m that were centered around the target. From the geometric theory of diffraction, we assume that a complex target can be decomposed into a sparse set of scattering centers. The scattering centers are then assumed to be $K$ point targets, and are described through using $\{(x_k, y_k), h_k(\theta, \phi)\}_{k=1}^K$, where $(x_k, y_k) \in [-\frac{L}{2}, \frac{L}{2}] \times [-\frac{L}{2}, \frac{L}{2}]$ are the spatial coordinates of the point targets, $\theta$ is the azimuthal angle, $\phi$ is the angle of elevation of the radar platform, and $h_k(\theta, \phi)$ is the corresponding scattering coefficients that depend on the viewing angle. The samples of the received signal after the standard de-chirping procedure are given by

$$s(f_m; \theta, \phi) = \sum_{k=1}^{K} h_k(\theta, \phi) \exp\left(-j4\pi \frac{f_m \cos(\phi)}{c}(x_k \cos(\theta) + y_k \sin(\theta))\right), \quad (2)$$

where $f_m$ is the illuminating frequencies such that $m \in [M]$, $M = \frac{2BL}{c}$; $B$ is the bandwidth of the transmitted pulse; $c$ is the speed of light; and the notation $[M]$ denotes the enumeration of natural numbers up till $M$. We estimated the function $h_k(\theta, \phi) \ \forall \ k \in [K]$ from the receiver samples.

Parametric models for standard reflectors, such as dihedral and trihedral reflectors, were studied in [71–73]. These models indicate that the reflectivity is a smooth function over the viewing angle, which is parameterized by the reflector's dimensions and orientation. Therefore, we exploited this smoothness to approximate this infinite-dimensional function through using interpolation strategies [74] with the available set of samples $\Theta$ in the angle

domain. We denote the sampled returns from the scene by the matrix $\mathbf{S} = NUFFT(X) \in \mathbb{C}^{N_\theta \times M}$, where NUFFT represents the non-uniform Fourier transform. The elements of $\mathbf{S}$ are defined as follows:

$$s_{m,i} = n_{m,i} + \sum_{k=1}^{K} h_k(\theta_i, \phi) \exp\left(-j4\pi \frac{f_m \cos(\phi)}{c}(x_k \cos(\theta_i) + y_k \sin(\theta_i))\right). \tag{3}$$

where $n_{m,i}$ represents the measurement noise. In order to solve the estimation problem, we assume that the function $h_k$ has a representation in the basis set denoted by the matrix $\mathbf{\Psi} \in \mathbb{C}^{N_\theta \times D}$ of a size $D$. For the MSTAR data set, the elevation angles we worked with are similar. We assumed that the variation in $h_k$ with respect to $\phi$ was insignificant. This assumption lead to the following relation $h_k(\theta; \phi) = \sum_{v=1}^{D} c_{v,k} \psi_v(\theta) + \epsilon_P$. The estimated phase history matrix was now $\hat{\mathbf{S}}$, whose elements were given by

$$\hat{s}_{m,i} = \hat{n}_{m,i} + \sum_{k=1}^{K} \sum_{v=1}^{D} c_{v,k} \psi_v(\theta_i) \exp\left(-j4\pi \frac{f_m \cos(\phi)}{c}(x_k \cos(\theta_i) + y_k \sin(\theta_i))\right), \tag{4}$$

where $\hat{n}_{m,i}$ consists of the measurement noise and the approximation error. To estimate the coefficients $c_{v,k}$ from the noisy measurements in (4), we discretized the scene with a resolution of $\Delta R$ in the $X, Y$ (range and cross-range, respectively) plane to obtain the $K = N_R^2$ grid points, where $N_R = \frac{2BL}{c}$ is the number of the range bins. Furthermore, we considered a smooth Gaussian function to perform the noisy interpolation. We partitioned the sub-aperture $2\Delta\theta$ into smaller intervals of equal length with a corresponding set containing the means of the intervals given by $\{\hat{\theta}_v\}_{v=1}^{D}$, where $D = 12$, and these were used as the centroids for the Gaussian interpolating functions. We assumed the width of the Gaussian function $\sigma_G$ as a constant hyper-parameter, whose selection is described in Section 3.4. Hence, $\sigma_G$ is the constant minimum persistence of the scattering center in the azimuth domain that we wish to detect. The radial basis functions used were

$$\psi_v(\theta) = \exp\left(-\left(\frac{\theta - \hat{\theta}_v}{2\sigma_G}\right)^2\right) \tag{5}$$

The elements of $\hat{\mathbf{S}}$ were obtained due to the scattering centers located at the discrete grid points, which are now given by

$$\hat{s}_{m,i} = \hat{n}_{m,i} + \sum_{k=1}^{N_R^2} \sum_{v=1}^{D} c_{v,k} \boldsymbol{\psi}_v(\theta_i) \exp\left(-j4\pi \frac{f_m \cos(\phi)}{c}(x_k \cos(\theta_i) + y_k \sin(\theta_i))\right). \tag{6}$$

Here, the discrete grids for $(x_k, y_k)$ and $(\theta_i, f_m)$ are now both known. Let the vectors containing all corresponding grid points for $x_k, y_k, \theta_i, and f_m$ be referred to as $\mathbf{x}, \mathbf{y}, \boldsymbol{\theta}, and \mathbf{f}$ respectively. The problem now is to find the coefficients $c_{v,k}$ that minimize the error between $\hat{\mathbf{S}}$ and $\mathbf{S}$. Let vector $\mathbf{c_k} = [c_{1,k} \cdots c_{D,k}]^T$. To recover the structured signal $\mathbf{h} = [h_1 \cdots h_{N_R^2}]$, which represents the scattering coefficient of a sparse scene that has a sparse representation in an underlying set of functions, we solve the following linear inverse problem via a sparse-group regularization on $\mathbf{c_k} \forall k \in [N_R^2]$.

$$\min_{\mathbf{C}} \left(\sum_{k=1}^{N_R^2} \lambda \|\mathbf{c}_k\|_2 + \|\mathbf{S} - \hat{\mathbf{S}}\|_F\right) \Longleftrightarrow \min_{\mathbf{C}} J(\mathbf{C}, \sigma_G) \tag{7}$$

where $\mathbf{C}$ refers to the matrix $[\mathbf{c_1} \cdots \mathbf{c_{N_R^2}}]$, $\sigma_G$ is a constant hyper-parameter, and $\|\cdot\|_2, \|\cdot\|_F$ refer to the $l^2$, Frobenius norms, respectively.

The elements of the recovered model, $\mathbf{S}^*(\boldsymbol{\theta}; \mathbf{f})$ are now

$$s^*_{i,m} = \sum_{k=1}^{N_R^2} \sum_{v=1}^{D} c^*_{v,k} \boldsymbol{\psi}_v(\theta_i) \exp\left(-j4\pi \frac{f_m \cos(\phi)}{c}(x_k \cos(\theta_i) + y_k \sin(\theta_i))\right) \quad (8)$$

where $c^*_{v,k}$ are the recovered coefficients. The phase history measurements were converted back to the image by using overlapping sub-apertures that spanned $2\Delta\theta = 3$ degrees in the azimuth domain, as shown in Figure 1. Here, $\Delta\theta$ is the angular span of the sub-aperture from the center azimuth. This azimuth span determines the cross-range resolution of the SAR image. The slant-plane cross-range resolution is given by

$$\Delta_{slant}^{CR} = \frac{\lambda_c}{4\sin(\Delta\Theta)},$$

$$\Delta\Theta = \sin^{-1}\left(\frac{\lambda_c}{4\Delta_{ground}^{CR}\cos(\phi)}\right).$$

Each image in the MSTAR data set contains a header that summarizes the imaging geometry information stored in the Phoenix format. The parameters, such as center-frequency $f_c$ and angle of depression $\phi$, are obtained from the header information stored in each file. We assume that the cross-range resolution given in the file is on the ground-plane. The cross-range resolution is given as $\Delta_{ground}^{CR} = 0.305$. We infer that $\Delta\Theta = 1.51$ degrees. We apply the same Taylor window with zero-padding and then translated it back to the Cartesian coordinates before applying the Fourier transform to generate the images to augment the data set.

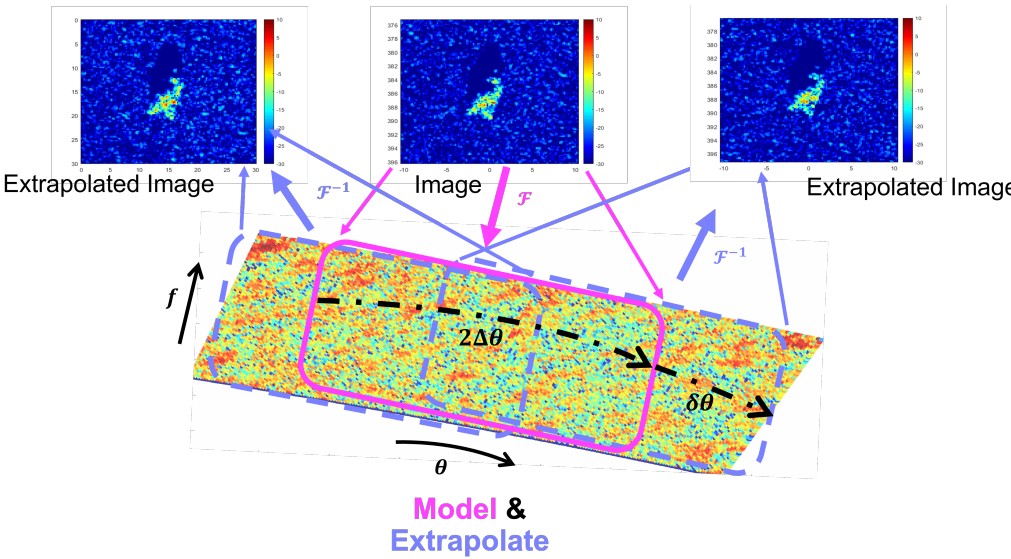

**Figure 1.** Pose synthesis using the phase history model. $\mathcal{F}$ denotes the Fourier transform operator. The phase history collected over an azimuth span of $2\Delta\theta = 3°$ was extrapolated by $\delta\theta$ via the model $\mathbf{S}^*(\boldsymbol{\theta}; \mathbf{f})$.

## 3. Experiments

We hypothesize that the underlying scattering mechanism is locally continuous or persistent in nearby look angles. We exploited this structure to generate realistic SAR images in the nearby look angles to augment the training data set. We evaluate that this hypothesis can be supported in the publicly available MSTAR data set, which has been typically used for evaluating algorithms.

### 3.1. MSTAR Data Set

One of the motivations for choosing the well-studied MSTAR data set for our experiments is its popularity, which enables us to compare our method with several other approaches in the literature. Additionally, imaging geometry parameters are provided in the MSTAR data set, which makes it straightforward to infer the frequency support of the images. Alternatively, for the data sets that do not provide the parameters, we can infer the frequency support of the image by applying Fourier transform to the complex-valued SAR image. The MSTAR data set consists of 10 classes, i.e., tanks (T62 and T72), armored vehicles (BRDM2, BMP2, BTR60, and BTR70), a rocket launcher (2S1), an air defense unit (ZSU234), a military truck (ZIL131), and a bulldozer (D7). We illustrate this observation in Figure 2.

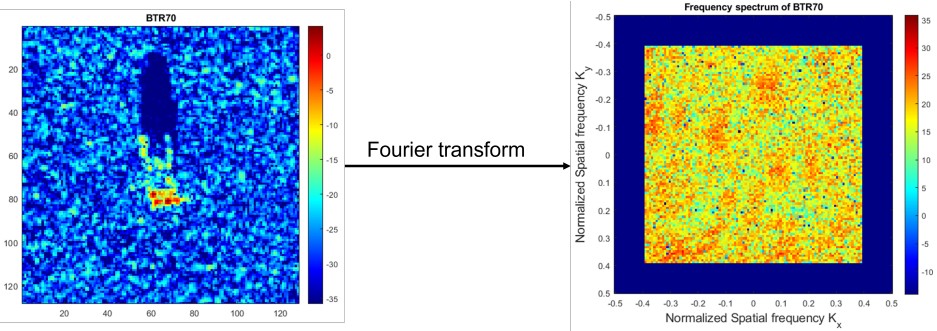

**Figure 2.** A $128 \times 128$ image chip from the MSTAR data set of a BTR-70 Tank and the corresponding spectral representation that was obtained by applying 2D Fourier transform to extract the K-space support.

We considered a image chip of BTR70 at the size of a $128 \times 128$ pixel-sized image, as well as the corresponding spectral content. The support exists on a central region of $100 \times 100$ frequency points. The corners of the square support can now be expressed in terms of the normalized spatial frequency domain, which is used to compute the K-space points. The phase history measurements at those K-space points were computed via the non-uniform Fourier transform [75,76] operator without a loss in generality. Therefore, we can estimate our model on this normalized domain by sampling in a wedge-shaped region. The radar platform used in constructing the MSTAR data set acquires the measurements through using $N_p = 100$ pulses over an aperture of 3 degrees. The phase history measurements obtained in the receiver were converted to images via the sub-aperture-based method described in [77]. The motion-compensation steps were followed by the application of a Taylor window to control the side-lobes. The measurements were zero-padded to obtain an over-sampled image via the Fourier transform. The complete MSTAR data set used in [15] was highly imbalanced. We replaced the data set used in [15] with a balanced subset, which is referred to as the standard operating conditions that were considered in [14,26]. We henceforth denote this subset as the SOC MSTAR data set.

Similar to the existing literature, we used the images at a depression angle of $\phi = 17°$ for training, while images at $\phi = 15°$ formed the test set. Similar to [15], we cropped the images to $64 \times 64$ with the objects in the center. Note that we cropped the images right before feeding it to the ANN. We performed the modeling and augmentation steps on the original images. Since our paper's objective is to investigate the effects of data augmentation, we worked with much smaller training data sets by artificially reducing the size of our data set to $\phi = 17°$. We exponentially sub-sampled by extracting only the $\mathcal{R}$ ratio of the samples from each class, where $\mathcal{R} \in \{2^{-5}, 2^{-4}, 2^{-3}, 2^{-2}, 2^{-1}, 2^0\}$. We ensured that the extracted images were uniformly distributed over the $[0, 2\pi]$ azimuthal angle domain for each sub-sampling ratio. This sub-sampling strategy is essential for ensuring that the learning algorithm obtains a complete view of the vehicle's scattering behavior. We further selected 15% of the uniformly distributed samples from this uniformly sub-sampled data as

the validation set, we then utilized the remaining 85% as the new training set. The training data $\mathcal{D}_{train}$ include a flip augmentation along the cross-range domain [15], as well as real-time translations along both the range and cross-range domains. These translations (in no. of pixels) are randomly sampled from the set $\{-6, -4, -2, 0, 2, 4, 6\}$ at every epoch; as such, $\mathcal{D}_{train}$ will be henceforth referred to as just the baseline data. We also included the flip augmentation in the final validation set $\mathcal{D}_{val}$, and no augmentations were included in the final test-set $\mathcal{D}_{test}$. We formed our $T(\mathcal{D}_{train})$ by performing the proposed pose augmentation on each radar image in the training data set, as described in Section 2.2. Additionally, our net transformation $T$ also includes sub-pixel level translations, as well as real-time pixel-level translations [50] in the range and cross-range domains. We used the sub-pixel shifts of the $\frac{1}{2}$ pixel, which corresponded to an approximately 0.15 m displacement in the Y-direction (range) as well as in the X-direction (cross-range) of the scene, where each pixel corresponds to 0.3 m in the range and cross-range domains.

*3.2. Network Architecture*

Our data augmentation algorithm was decoupled from the network architecture by realizing the ATR algorithm. For our experiments, we choose the simple CNN network architecture that was inspired by [15] and which is shown in Figure 3. We made such an architectural choice because, similar to Zhong and Ettinger [15], we wished to show that the classification performance of even simple CNN architectures can be improved through successful regularization through using domain-specific data augmentation for (in our case) SAR ATR. We modified the network and used batch-normalization layers after the ReLU activation in the convolutional layers. We deferred the use of dropout in the convolutional layers since batch normalization regularizes the optimization procedure [78]. After the last convolutional layer, we flattened out all the feature values and used a fully connected (FC) layer, which was followed by a dropout layer that was used to obtain the final set of features. These features were used to estimate class $Y_1$ of the input SAR images via training that was achieved by using the categorical cross-entropy loss function $\mathcal{L}_1$. We further modified the cosine loss, which is used for pose awareness in [15], to a pair of simpler losses by using the $Y_2 = \sin(\theta)$ and $Y_3 = \mathbb{1}_A(\theta)$ features, where $\theta$ is the azimuthal angle and $\mathbb{1}_A$ is the indicator function over set $A = [\frac{-\pi}{2}, \frac{\pi}{2}]$. The mean-squared-error loss $\mathcal{L}_2$ was used for training the network to estimate $Y_2$ and the binary cross-entropy loss $\mathcal{L}_3$ (which is used for training the network to estimate $Y_3$). These two features uniquely determined the azimuthal angle, and they remove the need for a cosine distance loss. In our experiments, while training the model, we found that this modification to the loss function resulted in improving the convergence of the optimization procedure. The loss function of $\mathcal{L}$ to find the network parameters is now

$$\mathcal{L}(w, \mathcal{D}) = \bar{\mathbb{E}}_{\mathcal{D}}[\mathcal{L}_1(w, X, Y_1) + \mathcal{L}_2(w, X, Y_2) + \mathcal{L}_3(w, X, Y_3)] \tag{9}$$

$$\mathcal{L}_1(w, X, Y_1) = -\sum_{p=1}^{10} Y_{1,p} \log(\hat{Y}_{1,p}(w, |X|))$$

$$\mathcal{L}_2(w, X, Y_2) = (Y_2 - \hat{Y}_2(w, |X|))^2$$

$$\mathcal{L}_3(w, X, Y_3) = -Y_3 \log(\hat{Y}_3(w, |X|)) - (1 - Y_3) \log(1 - \hat{Y}_3(w, |X|))$$

where $|.|$ denotes the absolute value, $X \in \mathbb{C}^{64 \times 64}, Y_1 \in \{0, 1\}^{10 \times 1}, Y_2 \in [-1, 1], and\ Y_3 \in \{0, 1\}$ refer to the complex radar images, the one-hot vector of the 10 classes, $\sin(\theta)$, and $\mathbb{1}_A(\theta)$, respectively. $\bar{\mathbb{E}}_{\mathcal{D}}$ refers to the empirical mean over data set $\mathcal{D}$, and $Y_{1,p}$ is the $p^{th}$ component of the vector $Y_1$. All the quantities withˆ(hat) are the corresponding estimates given by the ANN.

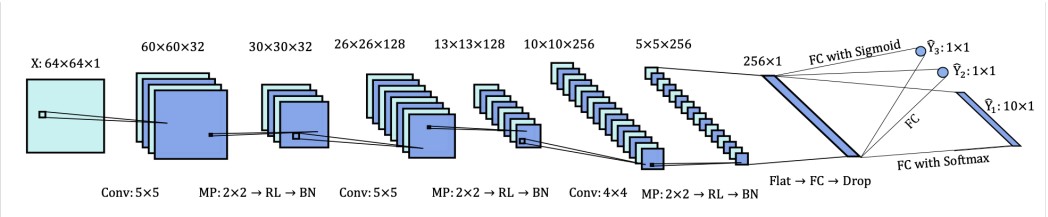

**Figure 3.** The neural network architecture. The abbreviations used are as follows. Conv is the convolutional layer followed by the kernel height × width. MP is the max pooling followed by the pooling size as the height × width. RL, BN, Flat, Drop, and FC are the ReLU, batch normalization, flattening, dropout and fully-connected layers, respectively. The sizes of the feature maps are mentioned at the top as height × width × channels.

### 3.3. Experimental Setup

The experiments were conducted using the network that is described in Section 3.2. This was used on the data sets described in Section 3.1. This model was trained on a local machine with a Titan Xp GPU. The Tensorflow (1.10) [79] library was used for its implementation through its Python API. We used the ReLU activation function everywhere except for in the final output layers of $\hat{Y}_1$, $\hat{Y}_2$, and $\hat{Y}_3$, where we instead used the Softmax, Linear, and Sigmoid activations, respectively.

An overview of the processing steps conducted to synthesize the radar images is as follows: Starting from the complex radar data, as described in Figure 4, we first transformed the image to a K-space by inverting the transformations applied to the MSTAR data in order to obtain the phase history representation. Through using the header information from the MSTAR data set, we determined the discrete grids for $(x_k, y_k)$ and $(\theta_i, f_m)$. Next, we estimated the model coefficients by solving the optimization problem described in Equation (7). As a result, we obtained the $\mathbf{S}^*(\boldsymbol{\theta}; \mathbf{f})$ model that is given by Equation (8). This model was further used to synthesize new columns of phase history data (or to extend the $\boldsymbol{\theta}$ vector). Consequently, a synthesized image was produced, which was achieved by the procedure described in Section 2.2 and followed by a transformation of the phase history data to complex-valued image data. The complete MATLAB code that was used to perform our proposed augmentations on the MSTAR data set is available at https://github.com/SENSE-Lab-OSU/mstar_data_aug (accessed on 11 July 2023).

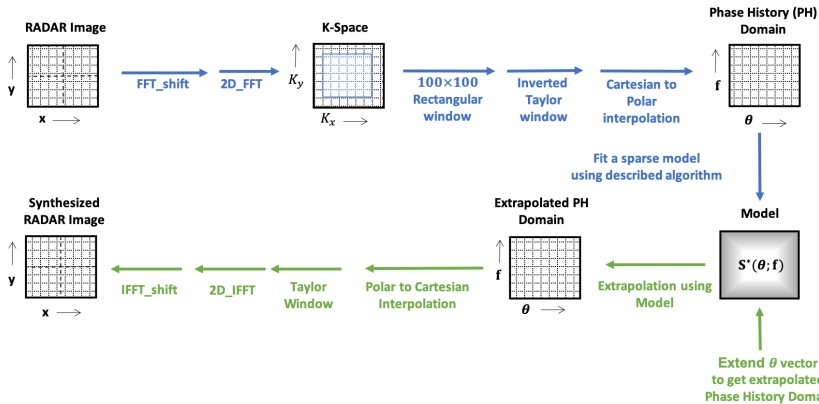

**Figure 4.** Overview of the Image Synthesizing Procedure. All boxes with grid lines represent the matrices of complex values across an $(x_k, y_k) \ \forall \ k \in [N_R^2]$ grid, where Cross-range and Range are labeled, as well as along an $(\theta_i, f_m) \ \forall \ i \in [N_\theta], m \in [M]$ grid, where $\boldsymbol{\theta}$ and $\mathbf{f}$ are also labeled. The blue arrows represent the pre-model fitting stage, and the green arrows represent the post-model fitting stage.

We used the magnitude of the complex-valued radar data as the input $X$, which is in agreement with the existing literature for training the network. We normalized all the input images to the unit norm to reduce some of the undesired effects that occur due to

the Gaussian kernel during extrapolation. We also removed all of the synthetic images at poses that were already in the corresponding training set. Then, the optimization problem in Equation (1) was solved by using the off-the-shelf method, as well as by the Adam variant of the mini-batch stochastic gradient descent optimizer with a mini-batch size of 64. The training was carried out for many epochs (>400) while using the early-stopping criterion, and the model was saved for the best moving average validation performance metric. Although we care about accuracy (i.e., that the percentage of samples are classified correctly) as a performance metric, the $\mathcal{D}_{val}$ here becomes small, especially for small $\mathcal{R}$ values, thereby saturating the validation accuracy at 100% and thus yielding this metric as less useful. Instead, we then monitored the minimum classification loss $\mathcal{L}_1$ as the validation performance metric. We reported the percentage error (or misclassification), which was $100 - $ accuracy, as the test performance results.

*3.4. Determining Hyper Parameters*

The PH model for each image and the neural network model introduced a set of hyper parameters. We will now explain our choices for a subset of them and will mention some others. The neural network's hyper parameters were kept at the Tensorflow (1.10) library's default values unless specified.

The PH model has two main hyper parameters, the $\sigma_G$ and $\delta\theta$. We determined, using a simple line-search, the optimum $\sigma_G$ for every image by minimizing the following equation over all possible values of it.

$$\sigma_G^* = \arg \min_{\sigma_G} \left[ \min_{\mathbf{C}} J(\mathbf{C}, \sigma_G) \right]$$

For determining the appropriate $\delta\theta$, we chose the heuristic approach for the grid search. We generated samples of up to $6°$ because the approximation error increases beyond that. We chose an appropriate $\delta\theta$ by running a grid search over a factor $\eta$, such that $\delta\theta = min\{6°, \eta\sigma_G^*\}$. This was chosen because the amount of possible extrapolation per image depends on the corresponding kernel width $\sigma_G^*$. We ran the training on the smallest subset of the data set at a sub-sampling ratio of $2^{-5}$ for the purpose of searching over a grid of three values, i.e., $\eta = \{1, 2, 3\}$. We chose $\eta = 3$ as it gives the best validation performance. Although we experimented with $\eta > 3$, we found the results were comparable to when $\eta = 3$.

For the neural network model, we set the dropout rate for the last fully connected layer at 0.2.

## 4. Results

A scarcity of training data affects the performance of the resulting ATR classifier in two distinct ways: First, a small training data set interferes with the ability of learning how to extract informative features from the data. Second, given a set of features, limited training data results in suboptimal decision boundaries, thereby leading to a poor generalization performance. We hypothesize that data augmentation techniques primarily improve the former effect, i.e., it improves the test performance through an enhanced training of the CNN's convolutional layers that serve as the feature extractors. Our empirical results, which are presented below, support this observation. With an adequate feature set, the classifier can be trained even with small training data sets, and it will still generalize well. To disentangle the two effects, the convolutional layers of the network were trained with the augmented training data set, and the classifier layers (after and including the first FC layer) were trained using the corresponding non-augmented training data set.

For all the approaches, we generated the results for the 6 values of $\mathcal{R}$ that correspond to the different sub-sampling ratios of the original training data set. All the models have the same architecture as described in Section 3.2, and they use the same $\mathcal{D}_{test}$. The difference among them is the $\mathcal{D}_{train}$ and $\mathcal{D}_{val}$ that are used, as described in Section 3.1. For consistency in the results, we repeated the process described in Section 3.1 to obtain four different

$\mathcal{D}_{train}$ and $\mathcal{D}_{val}$ for each $\mathcal{R}$ (except for $\mathcal{R} = 2^{-1}$ and $\mathcal{R} = 2^0$, which is where only two and one such unique data sets were possible, respectively). Moreover, we reported the mean and standard deviation of the classification performance. The overall augmentation performance is summarized in Tables 1 and 2, and they are visualized in Figure 5. The bold numbers in these tables highlight the best performance in respective rows.

**Table 1.** The test errors that correspond to the analysis plot (Figure 5a).

| Sub-Sampling Ratio ($\mathcal{R}$) | Baseline Data (B) | Adding Our Sub-Pixel Shifts (B + S) | Adding Our Poses and Sub-Pixel Shifts (B + S + P) | Full-Data Features (F) |
|---|---|---|---|---|
| $2^0$ | $0.50\ \{B_0\}$ | $0.58\ \{S_0\}$ | $\mathbf{0.37}\ \{SP_0\}$ | $0.50\ \{F_0\}$ |
| $2^{-1}$ | $1.44 \pm 0.33\ \{B_1\}$ | $1.03 \pm 0.08\ \{S_1\}$ | $\mathbf{0.54} \pm \mathbf{0.00}\ \{SP_1\}$ | $0.72 \pm 0.06\ \{F_1\}$ |
| $2^{-2}$ | $4.21 \pm 0.83\ \{B_2\}$ | $2.33 \pm 0.33\ \{S_2\}$ | $1.00 \pm 0.34\ \{SP_2\}$ | $\mathbf{0.99} \pm \mathbf{0.17}\ \{F_2\}$ |
| $2^{-3}$ | $10.99 \pm 0.73\ \{B_3\}$ | $5.68 \pm 0.67\ \{S_3\}$ | $3.32 \pm 1.31\ \{SP_3\}$ | $\mathbf{1.22} \pm \mathbf{0.26}\ \{F_3\}$ |
| $2^{-4}$ | $18.78 \pm 2.40\ \{B_4\}$ | $14.02 \pm 0.28\ \{S_4\}$ | $7.33 \pm 0.54\ \{SP_4\}$ | $\mathbf{2.07} \pm \mathbf{0.21}\ \{F_4\}$ |
| $2^{-5}$ | $32.38 \pm 2.93\ \{B_5\}$ | $29.98 \pm 2.62\ \{S_5\}$ | $18.66 \pm 3.22\ \{SP_5\}$ | $\mathbf{4.55} \pm \mathbf{0.57}\ \{F_5\}$ |

**Table 2.** Test Errors corresponding to the comparative plot (Figure 5b).

| Sub-Sampling Ratio ($\mathcal{R}$) | Baseline Data (B) | Augmenting with Naively Rotated Poses (B + R) | Augmenting with Linearly Interpolated Poses (B + L) | Augmenting with Our Poses and Sub-Pixel Shifts (B + S + P) |
|---|---|---|---|---|
| $2^0$ | $0.50\ \{B_0\}$ | $0.62\ \{R_0\}$ | $0.50\ \{L_0\}$ | $\mathbf{0.37}\ \{SP_0\}$ |
| $2^{-1}$ | $1.44 \pm 0.33\ \{B_1\}$ | $1.22 \pm 0.19\ \{R_1\}$ | $1.22 \pm 0.14\ \{L_1\}$ | $\mathbf{0.54} \pm \mathbf{0.00}\ \{SP_1\}$ |
| $2^{-2}$ | $4.21 \pm 0.83\ \{B_2\}$ | $4.38 \pm 0.72\ \{R_2\}$ | $2.56 \pm 0.21\ \{L_2\}$ | $\mathbf{1.00} \pm \mathbf{0.34}\ \{SP_2\}$ |
| $2^{-3}$ | $10.99 \pm 0.73\ \{B_3\}$ | $10.01 \pm 1.72\ \{R_3\}$ | $7.26 \pm 2.56\ \{L_3\}$ | $\mathbf{3.32} \pm \mathbf{1.31}\ \{SP_3\}$ |
| $2^{-4}$ | $18.78 \pm 2.40\ \{B_4\}$ | $19.83 \pm 2.21\ \{R_4\}$ | $12.99 \pm 1.10\ \{L_4\}$ | $\mathbf{7.33} \pm \mathbf{0.54}\ \{SP_4\}$ |
| $2^{-5}$ | $32.38 \pm 2.93\ \{B_5\}$ | $32.05 \pm 7.58\ \{R_5\}$ | $30.79 \pm 3.04\ \{L_5\}$ | $\mathbf{18.66} \pm \mathbf{3.22}\ \{SP_5\}$ |

### 4.1. Ablation Study of the Proposed Approach

We performed an ablation study of the two proposed augmentations, i.e., the sub-pixel and pose augmentations, by incrementally adding them to the baseline data. We abbreviated the data sets as follows: the baseline data as B (which includes the image domain flips and integer pixel translations); the baseline data with proposed sub-pixel augmentations as B + S; and the baseline data with the proposed sub-pixel and pose augmentations as B + S + P. Moreover, to provide a lower bound on the test error at all the sub-sampling ratios of the data-augmentation approaches, we used a genie-aided approach (non-realizable in practice) by utilizing the full SOC data set to learn the CNN features, but we still used only the sub-sampled training data-set for training the fully connected classifier layers. This formed the test-error curve, which is referred to as F (for full data) in Figure 5a.

The full-data plot in Figure 5a (values are in Table 1) shows the importance of extracting good quality features, i.e., if we had access to all the poses, we would learn very good features. Having good features makes classification quite easy, and this is evident from the low test errors that are found even in cases of very low data availability when learning the classifier. The sub-sampling had little effect on the generalization performance for the genie-aided case. The baseline data plot showed a considerable degree of test error, especially in cases of low training-data availability. This test error was reduced in the B + S data plot, as well as further reduced in the B + S + P data plot, which shows the effectiveness of both our strategies in improving the quality of features extracted by the CNN. Note that the majority of the improvement comes from the pose augmentations. For $\mathcal{R} = 2^{-5}$, the proposed augmentations reduced the test error by more than 42% when compared to the baseline approach (which includes the image domain flips and integer pixel translations). For $\mathcal{R} > 2^{-2}$, the model that used both proposed augmentations provided an even better performance than the genie-aided features. This makes sense because our augmentation strategy is able to successfully fill in the pose information gaps in the complete SOC data.

However, there exists a considerable gap between the full data and the B + S + P data plots in the smaller data regimes of $\mathcal{R} < 2^{-2}$. So, there still exists room for further improvements in aiding the network-learning informative features for the data-starved regimes.

The confusion matrices for a sample data set at $\mathcal{R} = 2^{-4}$ are shown in Tables 3 and 4. These tables clearly show that the performance has considerably improved via the proposed augmentation of the training data in cases of low data availability. Not only that, but the performance also improved over all the classes except two. As there were four distinct sub-sampled data sets at $\mathcal{R} = 2^{-4}$, we picked the one that was a good representative of the average performance. The bold numbers in these tables highlight the best performing method for respective classes.

*4.2. Comparison with Existing SAR-ATR Models*

In comparing our test error with the full SOC MSTAR data set, it can be seen from Table 5 that our approach is on par with the existing approaches when using all of the data. We are interested in training the CNN models when data availability is extremely low, say $\leq 60$ samples per class. To compare the results obtained from our approach to recent works in extremely low-data regimes, we utilized some results from [26,42]. We also conducted B + S + P experiments for 18% of data per class. These are also tabulated in Table 5. It is in this extreme sub-sampling regime where our approach outperforms all the other existing approaches. The proposed algorithm reduces the test error by more than 46% when compared to the next best approach of the CNN-TL-bypass [26]. We obtained the lowest test error even when using the smallest portion of the data. We reiterate that most of the tabulated approaches were decoupled from our data-augmentation approach. So, in principle, it may be possible to combine our data-augmentation strategy with the existing approaches in order to obtain even better results. The bold numbers in Table 5 highlight the best performing method in respective columns.

**Table 3.** Confusion matrix for the classifiers corresponding to $\mathcal{R} = 2^{-4}$ with no augmentation.

| Class | 2S1 | BMP2 | BRDM2 | BTR60 | BTR70 | D7 | T62 | T72 | ZIL131 | ZSU234 | Error (%) |
|---|---|---|---|---|---|---|---|---|---|---|---|
| 2S1 | 196 | 0 | 1 | 0 | 3 | 1 | 40 | 5 | 18 | 10 | 28.467 |
| BMP2 | 21 | 117 | 2 | 18 | 10 | 0 | 1 | 23 | 3 | 0 | 40.0 |
| BRDM2 | 9 | 1 | **256** | 1 | 0 | 1 | 0 | 0 | 6 | 0 | 6.569 |
| BTR60 | 2 | 2 | 4 | 161 | 10 | 3 | 2 | 4 | 4 | 3 | 17.436 |
| BTR70 | 21 | 13 | 1 | 23 | 130 | 1 | 0 | 6 | 0 | 1 | 33.673 |
| D7 | 0 | 0 | 0 | 0 | 0 | **264** | 1 | 0 | 7 | 2 | 3.65 |
| T62 | 5 | 0 | 0 | 2 | 0 | 1 | 234 | 4 | 22 | 5 | 14.286 |
| T72 | 5 | 2 | 0 | 3 | 0 | 1 | 16 | 164 | 5 | 0 | 16.327 |
| ZIL131 | 1 | 0 | 0 | 0 | 0 | 34 | 4 | 0 | 234 | 1 | 14.599 |
| ZSU234 | 0 | 0 | 0 | 0 | 0 | 17 | 11 | 0 | 29 | 217 | 20.803 |
| | | | | | Overall | | | | | | 18.639 |

**Table 4.** Confusion matrix for the classifier corresponding to $\mathcal{R} = 2^{-4}$ with augmentation.

| Class | 2S1 | BMP2 | BRDM2 | BTR60 | BTR70 | D7 | T62 | T72 | ZIL131 | ZSU234 | Error (%) |
|---|---|---|---|---|---|---|---|---|---|---|---|
| 2S1 | **251** | 0 | 0 | 1 | 0 | 1 | 9 | 8 | 4 | 0 | 8.394 |
| BMP2 | 4 | **169** | 0 | 4 | 0 | 0 | 4 | 12 | 1 | 1 | 13.333 |
| BRDM2 | 16 | 8 | 243 | 0 | 0 | 0 | 0 | 0 | 6 | 1 | 11.314 |
| BTR60 | 2 | 1 | 4 | **172** | 6 | 1 | 3 | 1 | 1 | 4 | 11.795 |
| BTR70 | 7 | 2 | 1 | 0 | **184** | 0 | 0 | 2 | 0 | 0 | 6.122 |
| D7 | 0 | 0 | 0 | 0 | 0 | 263 | 0 | 0 | 0 | 11 | 4.015 |
| T62 | 6 | 0 | 0 | 4 | 0 | 1 | **257** | 4 | 1 | 0 | 5.861 |
| T72 | 1 | 0 | 0 | 0 | 0 | 0 | 10 | **183** | 0 | 2 | 6.633 |
| ZIL131 | 6 | 0 | 0 | 0 | 0 | 8 | 6 | 1 | **244** | 9 | 10.949 |
| ZSU234 | 0 | 0 | 0 | 0 | 0 | 0 | 2 | 0 | 0 | **272** | 0.73 |
| | | | | | Overall | | | | | | **7.711** |

**Table 5.** Results from the various SAR-ATR efforts that used the SOC MSTAR data set.

| Method | Error (%) Using 100% Data | Error (%) Using ≤20% Data |
|---|---|---|
| SVM (2016) [40] | 13.27 | 47.75 (at 20%) |
| SRC (2016) [40] | 10.24 | 36.35 (at 20%) |
| A-ConvNet (2016) [14] | 0.87 | 35.90 (at 20%) |
| Ensemble DCHUN (2017) [43] | 0.91 | 25.94 (at 20%) |
| CNN-TL-bypass (2017) [26] | 0.91 | 2.85 (at 18%) |
| ResNet (2018) [44] | 0.33 | 5.70 (at 20%) |
| DFFN (2019) [42] | **0.17** | 7.71 (at 20%) |
| Our Method | 0.37 | **1.53** (at 18%) |

Except for the works of [50,59], we did not find reproducible data-augmentation strategies that explicitly synthesize samples at new poses. As pointed out earlier, our approach can be used in conjunction with most of the other strategies outlined in Section 1.1. As such, we conducted a detailed comparison of our pose synthesis approach and sub-pixel level translations with the pose synthesis methods in [50,59]. We added real-time pixel level translations for all experiments. For the sake of completion, the simple rotations that are produced in [59] used the rotation matrix and were followed by appropriate cropping, and the linearly interpolated poses that were synthesized in [50] used the following equation

$$I_{\theta_c} = CR_{\theta_c}\left( \frac{|\theta_b - \theta_c|R_{\theta_a}(I_{\theta_a}) + |\theta_a - \theta_c|R_{\theta_b}(I_{\theta_b})}{|\theta_a - \theta_c| + |\theta_b - \theta_c|} \right) \tag{10}$$

where $R_\theta(I)$ denotes the rotation of the radar image $I$ by $\theta$ degrees clockwise, $CR_\theta(I)$ denotes the same but counter-clockwise, $I_\theta$ denotes the radar image at the pose $\theta$, $\theta_c$ is the desired new pose, and $\theta_a$ and $\theta_b$ are the poses closest to $\theta_c$ in the training data.

For a qualitative evaluation, we illustrated the images that were synthesized from the *T62* tank at $\theta_c = 57°$, and the corresponding ground-truth image (which is part of the Full SOC data) was used for comparison purposes only. We observed in Figure 6b the synthesized image at $\theta_c = 57°$ when using $\theta_a = 56°$ and $\theta_b = 85°$, and this was achieved by using a sub-sampled data set with $\mathcal{R} = 2^{-4}$. We note that the dominant scattering centers in the synthesized image were different compared to the ground-truth image. Next, we used the proposed model that was estimated for the azimuth angle of 56°. It is evident from Figure 6c that the synthesized image from our method captures all the dominant scattering centers present in the ground truth. Finally, we used the rotation operator to synthesize at $\theta_c = 57°$ when using $\theta = 56°$. We observed that, even when the rotation is small, the ground-truth image has a different scattering behavior that is not captured by rotation.

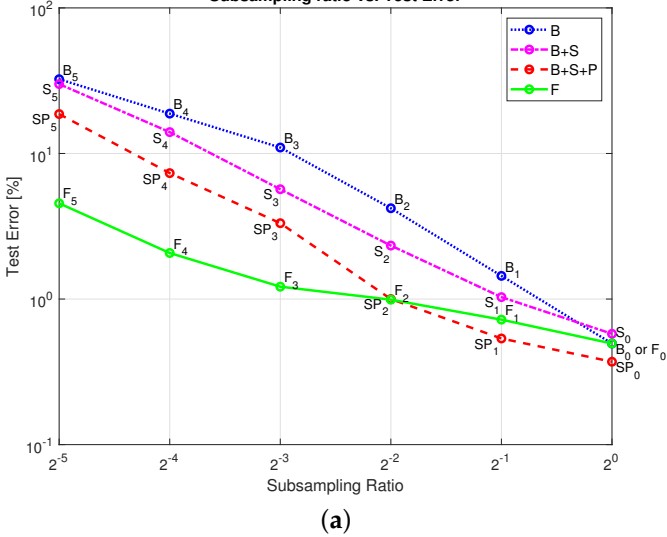

(a)

**Figure 5.** *Cont.*

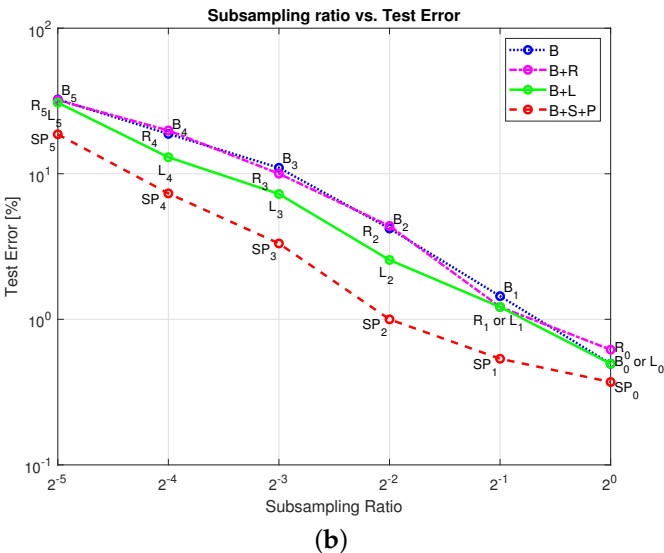

**Figure 5.** Quantitative evaluation of the proposed approach. (**a**) Ablation study. (**b**) Comparison with other augmentations.

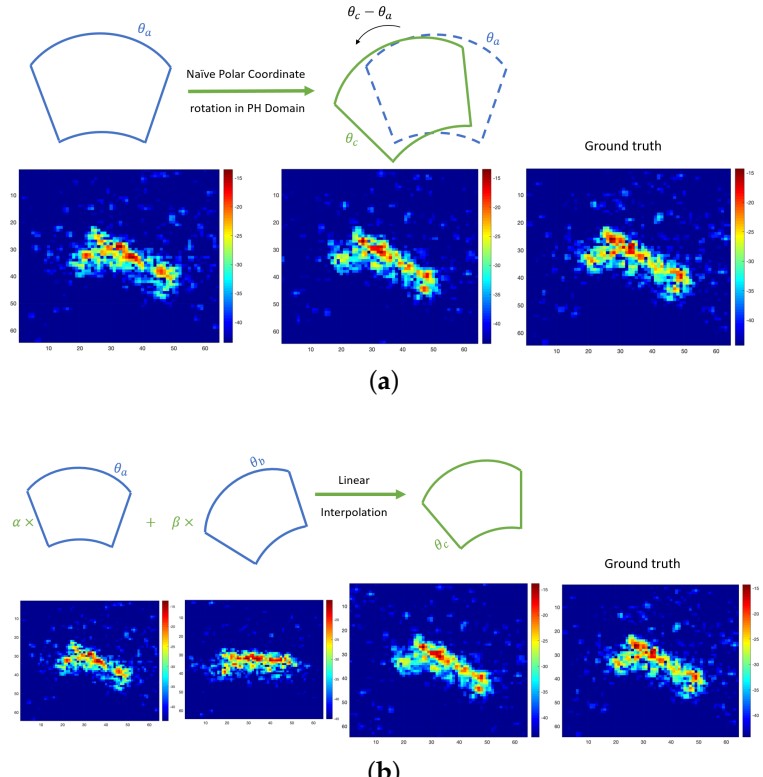

**Figure 6.** *Cont.*

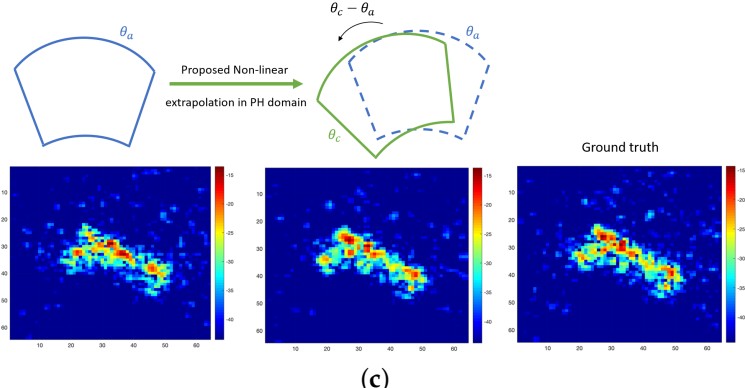

**(c)**

**Figure 6.** Comparison of the radar images synthesized for class T62 at the viewing angle of $\theta_c = 57°$ when using different augmentation strategies. (**a**) Radar image at a viewing angle of $\theta_c = 57°$, which was generated by rotating the closest available in the sub-sampled data set at $\theta_a = 56°$ (from [59]). (**b**) Linear interpolation strategy proposed in [50]. Here, $\alpha$ and $\beta$ can be inferred from Equation (10), and the closest poses to $\theta_c = 57°$ in the sub-sampled data set were $\theta_a = 56°$ and $\theta_b = 85°$. (**c**) This radar image at the azimuth angle of $\theta_a = 56°$ was first approximated using the proposed set of basis functions in the frequency domain. Measurements from the unobserved viewing angles were synthesized via the model in the frequency domain, which was used to create the image at the viewing angle of $\theta_c = 57°$.

For the quantitative evaluation, we compared the ATR performance at all sub-sampling ratios similar to those conducted in the previous Section 4.1. We abbreviated the data sets as follows: the baseline data as B, the baseline data with simple rotations added (from [59]) as B + R, the baseline data with linearly interpolated poses (from [50]) as B + L, and the baseline data with proposed sub-pixel and pose augmentations as B + S + P.

The comparison of these is tabulated in Table 2, and it can also be seen in Figure 5b. It is evident that our approach is significantly better than both simple rotations and linearly interpolated poses for this CNN-based ATR task. For $\mathcal{R} = 2^{-5}$, the proposed augmentations reduced the test error by more than 39% when compared to the next best augmentation approach of [50].

## 5. Conclusions and Future Directions

In this paper, we proposed incorporating the domain knowledge of SAR phenomenology into a CNN by way of data augmentation. We presented a model-based approach to data augmentation for the purpose of training the neural network architecture to solve the ATR problem with limited labeled data. Through extensive simulation studies, we showed the effectiveness of the augmentation strategies by training a neural network with the augmented data set that was synthesized from the phase history models extracted from each available training image. Our results show that the proposed data augmentation strategy produced a significant improvement in the model's generalization performance when compared to the baseline performance over a wide range of sub-sampling ratios. As presented, the phase history approximation method is only valid in a local neighborhood of a given azimuth angle. Future work could focus on fitting a single global model to every class that is jointly derived from all the training images. Such a global model could produce a diverse set of SAR images over larger pose variations. Since, typically, target image chips are not perfectly registered and are aligned across different azimuth angles, the global model fit should incorporate unknown phase and spatial shifts for each image. As part of future research, we propose developing a network architecture to support a unified model that can account for these phase errors, and which can synthesize a larger data set to improve the classifier's performance further. We also aim to evaluate such augmentations on more challenging data sets such as the Military Ground Targets Database (MGTD) [80], the Synthetic and Measured Paired Labeled Experiment (SAMPLE) [81], and the Ship data

set OpenSARShip (which was obtained from Sentinel-I imagery [82]). Additionally, since our data-augmentation method creates complex-valued synthetic data, it can potentially be used to regularize complex-valued neural networks [83,84], and it can be used to improve their performance for SAR ATR. Moreover, since the currently used Taylor windowing is sub-optimal, the problem in optimizing the window function that enhances ATR performance can be incorporated in the form of a multi-objective optimization—a topic that will be investigated in our future work.

**Author Contributions:** Conceptualization, T.A., N.S. and E.E.; methodology, T.A., N.S. and E.E.; software, T.A. and N.S.; validation, T.A. and N.S.; formal analysis, T.A. and N.S.; investigation, T.A.; resources, E.E.; data curation, N.S.; writing—original draft preparation, T.A.; writing—review and editing, N.S. and E.E.; visualization, T.A. and N.S.; supervision, E.E.; project administration, E.E.; funding acquisition, E.E. All authors have read and agreed to the published version of the manuscript.

**Funding:** This research was partially supported by the Army Research Office (grant W911NF-11-1-0391 and NSF Grant IIS-1231577).

**Data Availability Statement:** The publicly available MSTAR data set was used in this study. These data can be obtained from the US Air Force Research Laboratory (AFRL) website https://www.sdms.afrl.af.mil/index.php?collection=mstar (accessed on 11 July 2023). The data pre-processing conducted in this study is described in Section 3.1.

**Conflicts of Interest:** The authors declare no conflict of interest. The funders had no role in the design of the study; in the collection, analyses, or interpretation of data; in the writing of the manuscript; or in the decision to publish the results.

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
