# Peer review of "Sparse Signal Models for Data Augmentation in Deep Learning ATR"

_remotesensing, doi:10.3390/rs15164109_

Round 1

Reviewer 1 Report

This paper  proposed a data augmentation method of limited dataset using a principled approach by exploiting the phenomenology of the RF backscatter data.  And this work is an extension of their  previous  work.

Nice work.  But there are some issues to be addressed.

1.  In the related work, some works using multiple polarization information to enhance SAR ATR precision should be introduced.  (For example, A Dual-Polarization Information-Guided Network for SAR Ship Classification).

2.  In Figure 3, the neural network architecture used in the experiments is as simple as some CNNs in 2015.  But it is 2023 already.   RestNet-50 or RestNet-101 are commonly used backbone in classification task.  The author should clarify the reason why they choose such simple CNN architecture in experiments.

3.  MSTAR seems to be well studied and the accuracy of state-of-the-art models in MSTAR are very high.  Are there more challenging datasets of SAR ATR to be used to verify the proposed method in future work?

Reviewer 2 Report

1. This is an excellent paper that significantly improves ATR accuracy for SAR; it is very clear and thorough, and it introduces several new good ideas.

2. The authors should consider using complex-valued data as inputs to the neural net, as well as modified ADAM for complex-valued data and modified ReLU for complex-valued data; see the recent paper: "A Fair Performance Comparison between Complex-Valued and Real-Valued Neural Networks for Disease Detection," by Mario Jojoa, Begonya Garcia-Zapirain and Winston Percybrooks, MDPI August 2022, as well as Sarroff, A. Complex Neural Networks for Audio. Ph.D. Thesis, Dartmouth College, Hanover, NH, USA, May 2018.  https://digitalcommons.dartmouth.edu/dissertations/55.  The authors should also look at more recent work on complex-valued neural nets (CVNN).

3. The neural net architecture was not really the state-of-the-art for this type of application.  The authors might consider RESNET or NODE or more recent advances (e.g., see recent NIPS conferences).

4. Taylor weighting is suboptimal, and the authors might consider better methods.

Round 2

Reviewer 1 Report

Nice work. Thanks for your contribution.